# Health-related quality of life in patients with implantable cardioverter defibrillators in Sweden: a cross-sectional observational trial

Peter Magnusson ![ORCID],[1,2,3] Gustav Mattsson,[2] Marita Wallhagen,[4] Jan Karlsson[5]

For numbered affiliations see end of article.

**Correspondence to**
Dr Peter Magnusson;
peter.magnusson@
regiongavleborg.se

## ABSTRACT

**Objectives** Decisions regarding implantable cardioverter defibrillators (ICDs) must consider information about presumed health-related quality of life (HRQL). The purpose of the study was to assess HRQL in patients with ICD and compare it to a Swedish age-matched and sex-matched population.

**Design** Cross-sectional observational trial.

**Setting** Swedish ICD cohort.

**Interventions** Short form 36 (SF-36) questionnaires from ICD recipients implanted 2007–2017 (response rate 77.2%) were analysed using Mann-Whitney U test and effect size (ES).

**Results** In total, 223 patients (mean age 71.1±9.7 years, 82.1% men) were included. In most SF-36 domains (physical functioning (PF), role physical, general health (GH), vitality, social functioning and mental health), the score for patients with ICD was significantly lower (ES range 0.23–0.41, ie, small difference) than norms, except for bodily pain and role emotional. Both the physical component summary (PCS) and the mental component summary (MCS) scores had ES=0.31. Men and women had similar scores. Primary and secondary prevention patients scored similarly, except for worse GH in primary prevention (p=0.016, ES=0.35). Atrial fibrillation was associated with worse PF (ES=0.41) and PCS (ES=0.38). Appropriate therapy, inappropriate shock or complications requiring surgery were not associated with lower scores in any domain. In primary prevention due to ischaemic versus non-ischaemic cardiomyopathy, no domain was significantly different. PCS decreased with higher age strata (p=0.002) in contrast to MCS (p=0.986).

**Conclusions** Patients with ICDs have lower physical and mental HRQL than age-matched and sex-matched norms; however, the ESs are small. HRQL is similar regardless of sex, primary/secondary prevention indication, appropriate therapy, inappropriate shock or complications, but decreases with advancing age.

## INTRODUCTION

An implantable cardioverter defibrillator (ICD) effectively terminates ventricular arrhythmias and protects from severe bradycardia. The ICD is a cornerstone in the treatment of heart failure (HF) with reduced ejection fraction but is also used for miscellaneous other conditions and is generally cost-effective.[1–3] In addition to long-term protection from sudden cardiac death, ICDs can reduce symptoms of HF and prolong life when combined with cardiac resynchronisation therapy (CRT).[2] In survivors of ventricular fibrillation or sustained ventricular arrhythmias with haemodynamic compromise, secondary prevention is an established indication, providing a reasonable life expectancy with good functional status for more than 1 year.[1 4] Using similar reasoning regarding life expectancy, primary prevention in high-risk patients is recommended.[1 4] Unfortunately, adverse events related to ICDs occur; the cumulative incidence at 12 years is about 20% inappropriate shock, 6% device-related infections and 17% lead failure, and the incidence of complications seems to be underestimated in reports from registries.[5 6] The ICD affects several aspects of life, both professional life and leisure time activities. There may be restrictions related to driving, sports activities, and there is a risk for electromagnetic interference. ICD treatment has been associated with sleep disturbances, body perception concerns, anxiety and depression.

Patient-reported outcome measurements, such as health-related quality of life (HRQL),

is recognised as a valuable research tool. It contributes information beyond echocardiographically assessed ejection fraction, biomarkers, or physician-assessed New York Heart Association (NYHA) functional class.[7] [8] Several studies, mostly from controlled trials with interventions, have evaluated HRQL in ICD cohorts. The assessment of HRQL can be measured using a generic and/or a disease-specific questionnaire. The widely used short form 36 (SF-36) often provides a basis for generic evaluation in diverse populations.[9] [10] Whereas, worsening HRQL has been shown in numerous ICD cohorts, it remains largely unknown if this can be generalised into unselected ICD cohorts with long-term follow-up outside of a clinical trial.[11] [12] Therefore, the primary aim of this study was to determine SF-36 scores in unselected patients with ICD in comparison with age-matched and sex-matched general Swedish population norms. The secondary aims were to assess HRQL with regard to baseline atrial fibrillation (AF) or HF, appropriate ICD therapy, inappropriate ICD shock, device-related complications and compare secondary versus primary indications.

### Strengths and limitations of this study

► Real-world data on health-related quality of life (HRQL) in unselected implantable cardioverter defibrillator (ICD) patients.

► Long-term ICD treatment, with associations of ICD outcomes and subgroup analyses.

► Generic health-related quality of life using short form 36 compared with age-matched and sex-matched population norms.

► Cross-sectional data without longitudinal assessment of health-related quality of life.

► Lack of ICD-specific patient-reported questions.

## METHODS
### Design, setting and selection
This retrospective observational study covered adult patients in Region Gävleborg, Sweden, who had an ICD implanted or underwent device replacement between 1 January 2007 and 1 January 2017. Eligible patients were identified through Provisio, a software used for scheduling surgeries, with complete coverage of all device implants. The results from this cohort has been published previously and cover long-term follow-up with regard to appropriate therapy, complications and mortality.[13]

Data were retrieved from electronic medical records (Melior, Cerner Sverige AB, Stockholm) between March 2017 and February 2018 and evaluated according to a predefined protocol and imported from Excel 2010 (Microsoft Corporation, Redmond, Washington) into SPSS V.22 (IBM) and Stata (StataCorp (2017), Stata Statistical Software: Release 15, College Station, Texas) for statistical analyses. The questionnaires were distributed by regular mail, including two reminders, during 2019.

### Statistical analyses
Data were described as frequencies, percentages, means including SD and 95% CIs were used. The t-test was used for comparisons of continuous variables. Differences in SF-36 domains between groups were tested from the non-parametric Mann-Whitney U test. The associations between age and HQRL were tested using Spearman's non-parametric correlation coefficient for analyses and Pearson's correlation analysis to confirm results. Age was categorised into the four strata: 32–59 years, 60–69 years, 70–79 years and ≥80 years, and differences between age strata were analysed using the Kruskal-Wallis non-parametric analysis of variance. The estimation of the magnitude of difference between group was further assessed by calculating effect sizes (ESs), that is, the mean difference, divided by the pooled standard deviation (Cohen's d). The interpretation of ES was described according to standard criteria: trivial (<0.20), small (0.20–0.49), moderate (0.50–0.79) and large (≥0.80).[14] All two-sided p values<0.05 were considered statistically significant. The software programs Excel 2010 (Microsoft Corporation, Redmond, Washington), IBM SPSS Statistics for Windows, V.22.0: IBM, and SAS V.9.2 (SAS Institute) was used for analyses.

### Definitions of variables
The baseline clinical characteristics of the patients were retrieved from medical records by physicians under supervision of an expert in the field (PM).

ICD therapy was defined as appropriate if an event of either antitachycardia pacing or capacitor discharge of 30–45 J due to an episode of ventricular arrhythmia above the programmed detection interval and duration. ICD shock was classified as inappropriate if it was due to false detection or interpretation of myocardial signals (supraventricular tachycardia, T-wave oversensing) or signals outside the heart (caused by lead defect, myopotentials, electromagnetic interference).

Primary indication refers to patients who underwent ICD implant based on risk factors without any known secondary indication, that is, survival of documented sustained ventricular arrhythmia with haemodynamic compromise including ventricular fibrillation.

AF or flutter had to be at least 30 s in duration and documented by 12-lead ECG. HF classification was based on systolic dysfunction with an ejection fraction ≤50%. The concept 'complication' was restricted to events that required a surgical intervention related to the device system or the implant procedure, but excluded elective device replacement.

### SF-36 domains and validation
The HRQL was assessed by the SF-36 health survey version 1.[15] This is a patient-reported multidimensional HRQL tool derived from the Medical Outcome Study in the 1990s.[10] It has been widely used for determination of the patient-perceived burden of different medical conditions and is considered a generic instrument. The instrument has 36 items, which measures eight domains that cover both physical and mental health (MH) domains: physical functioning (PF), role-physical (RP), bodily pain (BP), general health (GH), vitality (VT), social functioning (SF), role-emotional (RE), and MH. In each domain, the scores range from 0 to 100 and higher values indicate better HRQL. Based on these domains, the physical component summary (PCS) and the mental component summary (MCS) can be calculated using norm-based scoring with a mean of 50. A score above 50 would be interpreted as better HRQL than the general Swedish population. The validation of the Swedish SF-36 yielded Cronbach's α between 0.79 (RE) and 0.93 (BP) for the domains.[10]

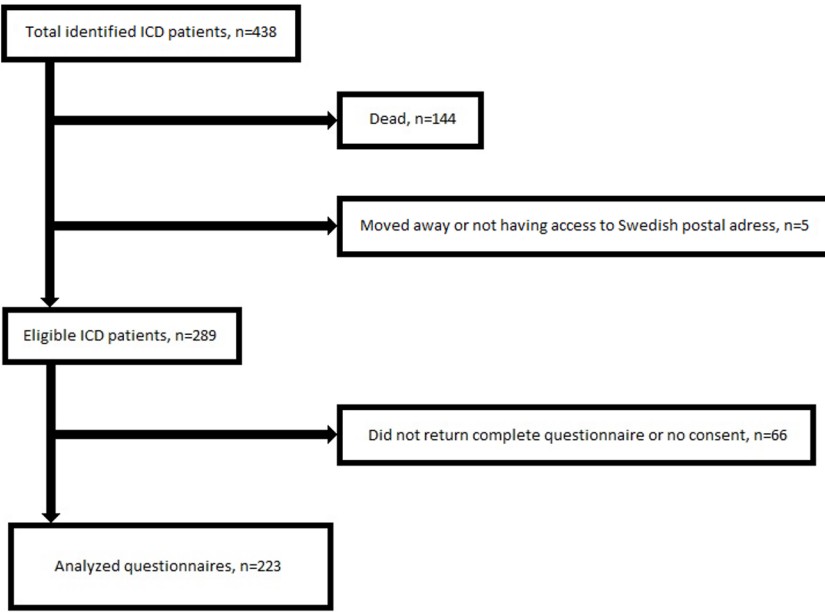

**Figure 1** Flowchart. ICD, implantable cardioverter defibrillator.

The SF-36 profile in the study group was compared with a general population sample which was randomly selected from the Swedish SF-36 normative database (n=8930; response rate 68%).[15] The normative sample (validated in Sweden 1991–1992) was matched on sex and age and comprised 171 persons (141 men) with a mean age of 68.8 years (SD 11.1).

### Patient and public involvement
Patients were not involved in the design, or conduct, or reporting, or dissemination plan of our research.

### RESULTS
### Cohort characteristics and follow-up
The SF-36 questionnaire was returned by 223 patients with ICD that corresponds to a response rate of 77.2%. The mean age was similar between those who responded and those who did not (p=0.276). The same was seen between sexes (p=0.255). Figure 1 depicts the flowchart of the emerged cohort. The majority was men 82.1% (n=183) and 60.1% (n=134) had an ICD due to primary prevention. All leads in the device systems were implanted transvenously. The baseline characteristics of the patients are summarised in table 1.

A total of 52 patients (23.3%) experienced appropriate ICD therapy, whereas 14 (6.3%) experienced inappropriate ICD shock during the follow-up period. Complications that required surgical intervention occurred in 39 patients (17.5%). Among patients with cardiomyopathy, 76 had ischaemic disease and 56 had no known underlying ischaemic disease.

The ages ranged from 32 to 90 years with a median age of 71.2 years. The mean age was 71.1 SD 9.7 years. Men and women had similar mean ages (71.1 vs 70.9 years). Similarly, the mean ages were not different for primary

versus secondary prevention (70.3 vs 72.3 years; p=0.138), a history of appropriate ICD therapy (70.7 vs 72.3 years; p=0.286), inappropriate shock (71.1 vs 71.0 years; p=0.967) or a complication requiring surgery (71.1 vs 70.9 years; p=0.907). On the contrary, patients with ischaemic cardiomyopathy were older than non-ischaemic cardiomyopathy patients (72.9 vs 66.9 years; p<0.001) as

**Table 1** Characteristics at ICD implant of 223 patients who completed the SF-36

| Patients | 223 |
|---|---|
| Mean age at implant | 64.1±10.9 years |
| Mean age at SF-36 | 71.1±9.7 years |
| Females | 40 (17.9%) |
| Device type | |
| ICD-VR | 36 (16.1%) |
| ICD-DR | 112 (50.2%) |
| CRT-D | 75 (33.6%) |
| Hypertension | 104 (46.6%) |
| Diabetes mellitus | 47 (21.1%) |
| Renal failure* | 27 (12.1%) |
| Atrial fibrillation | 62 (27.8%) |
| Beta-blockers | 201 (90.1%) |
| ACE-i/ARB | 193 (86.5%) |
| MRA | 104 (46.6%) |

Data presented as frequencies (percentage in parenthesis).
*Defined as S-creatinine≥130 µmol/L.
ACE-i, ACE inhibitor; ARB, angiotensin receptor blockers; CRT-D, cardiac resynchronisation therapy defibrillator; ICD-DR, dual lead implantable cardioverter defibrillator; ICD-VR, single lead implantable cardioverter defibrillator; MRA, mineralcorticoid receptor antagonist; SF-36, short form 36.

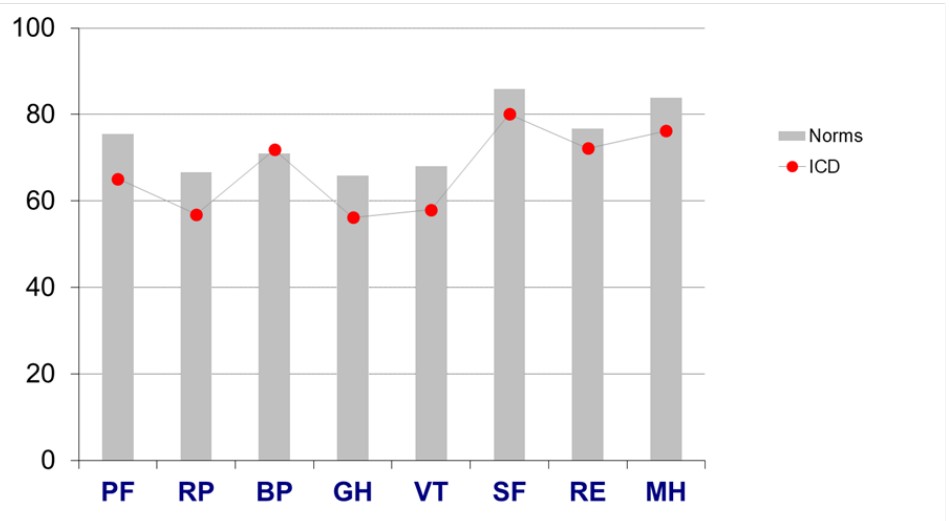

**Figure 2** Bar chart: Health-related quality of life among patients with ICD compared with age-matched and sex-matched norms. BP, bodily pain; GH,general health; ICD, implantable cardioverter defibrillator; MH, mental health; PF, physical functioning; RE, role-emotional; RP, role-physical; SF, social functioning; VT, vitality.

were patients with a history of AF at baseline (73.4 vs 70.1 years; p=0.024).

### HRQL in the ICD cohort compared with general population norms

In six out of eight SF-36 domains, there was a significantly lower score in patients with ICD compared with sex-matched and aged-matched population norms (figure 2, table 2).

There was a statistically significant (p<0.05) difference for PF, RP, GH, VT, SF and MH, but not for BP and RE. The ES among the significantly lower domains ranged from 0.23 (RP) to 0.41 (GH). Both component scores were significantly lower and the ESs similar, PCS (0.31) and MCS (0.31). Thus, all ESs were classified as small. Men and women reported similar SF-36 scores with non-significant differences in all domains including the component scores; PCS 41.0 SD 12.4 vs 39.5 SD

12.0 (p=0.406) and MCS 48.7 SD 10.9 vs 48.3 SD 10.5 (p=0.644).

Increasing age was associated with significantly lower scores on PF (r=−0.36; p<0.001), RP (r=−0.25; p<0.001), BP (r=−0.20; p=0.003), RE (r=−0.16; p=0.017) and PCS (r=−0.28; p<0.001).

SF-36 scores for the following age strata: 32–59, 60–69, 70–79 and ≥80 years, are reported in table 3. Kruskal-Wallis test showed significantly lower scores for PF, RP and PCS, while BP and RE showed a trend toward lower scores.

### HRQL subgroup analyses

The subgroup analyses for AF, complications, appropriate therapy and inappropriate shocks are reported in table 4. Patients with primary prevention ICDs were compared with secondary prevention indications. The age was similar, 70.3 vs 72.3 years. Primary prevention patients

**Table 2** SF-36 score in patients with ICDs compared with general Swedish population norms

| SF-36 domains | Cohort mean | SD | 95% CI | Effect size | Norm mean | SD | 95% CI | P value |
|---|---|---|---|---|---|---|---|---|
| Physical functioning | 65.1 | 28.1 | 61.4 to 68.8 | 0.38 | 75.5 | 26.0 | 71.5 to 79.6 | **0.0002** |
| Role physical | 56.9 | 44.1 | 50.9 to 62.8 | 0.23 | 66.7 | 40.4 | 60.2 to 73.2 | **0.0466** |
| Bodily pain | 71.9 | 29.9 | 67.9 to 75.8 | | 71.0 | 27.2 | 66.9 to 75.1 | 0.6395 |
| General health | 56.2 | 21.7 | 53.3 to 59.1 | 0.41 | 65.9 | 25.3 | 61.9 to 70.0 | **<0.0001** |
| Vitality | 58.0 | 24.8 | 54.7 to 61.3 | 0.39 | 68.1 | 27.2 | 63.7 to 72.4 | **<0.0001** |
| Social functioning | 80.1 | 24.4 | 76.8 to 83.3 | 0.25 | 85.9 | 22.9 | 82.4 to 89.4 | **0.0025** |
| Role emotional | 72.2 | 39.9 | 66.9 to 77.6 | | 76.7 | 35.7 | 71.1 to 82.4 | 0.3730 |
| Mental health | 76.2 | 19.4 | 73.6 to 78.8 | 0.38 | 83.8 | 20.9 | 80.5 to 87.2 | **<0.0001** |
| Physical component summary | 40.7 | 12.3 | 39.0 to 42.4 | 0.31 | 44.4 | 11.6 | 42.4 to 46.4 | **0.0052** |
| Mental component summary | 48.6 | 10.8 | 47.2 to 50.1 | 0.31 | 51.9 | 10.5 | 50.1 to 53.8 | **0.0010** |

Bold P values refer to significant values (<0.05).
ICD, implantable cardioverter defibrillator; SF-36, short form 36.

**Table 3** Short form 36 (SF-36) score in patients with implantable cardioverter defibrillators stratified by age

| SF-36 domains | 32–59 (n=25) Mean (SD) | 60–69 (n=72) Mean (SD) | 70–79 (n=76) Mean (SD) | 80+ (n=50) Mean (SD) | P value* |
|---|---|---|---|---|---|
| Physical functioning | 79.4 (25.4) | 71.9 (26.3) | 62.7 (29.0) | 51.4 (24.4) | **<0.0001** |
| Role-physical | 80.0 (36.8) | 60.0 (44.9) | 56.0 (43.9) | 40.9 (41.6) | **0.0050** |
| Bodily pain | 80.8 (29.3) | 74.4 (30.9) | 71.5 (28.8) | 64.2 (29.2) | 0.0780 |
| General health | 57.5 (21.2) | 56.0 (22.6) | 56.2 (21.9) | 55.7 (20.7) | 0.9827 |
| Vitality | 64.6 (20.4) | 59.7 (27.5) | 56.5 (23.8) | 54.3 (23.9) | 0.2363 |
| Social functioning | 84.0 (22.7) | 79.3 (27.9) | 80.3 (22.1) | 78.8 (23.5) | 0.8341 |
| Role-emotional | 82.7 (33.5) | 75.6 (38.6) | 73.6 (39.1) | 59.7 (44.0) | 0.0773 |
| Mental health | 77.8 (18.2) | 77.2 (21.2) | 74.4 (18.8) | 76.8 (18.3) | 0.6190 |
| Physical component summary | 46.6 (11.1) | 42.2 (12.7) | 40.0 (11.9) | 36.0 (11.4) | **0.0021** |
| Mental component summary | 48.6 (10.3) | 48.3 (12.0) | 48.6 (10.4) | 49.3 (10.1) | 0.9859 |

Bold P values refer to significant values (<0.05).

scored lower GH (p=0.016; ES 0.35) and there was a trend toward lower VT (p=0.097; ES=0.21), but otherwise no significant differences.

A history of AF at baseline was associated with significantly lower PF (p=0.004; ES=0.41) and a trend toward lower BP (p=0.083; ES=0.27) and VT (p=0.057; ES=0.31). This translated into lower PCS (p=0.013; ES 0.38) but similar MCS (p=0.928). However, the mean age of patients with AF was slightly higher (73.4 vs 70.1 years; p=0.024).

The mean age of patients who experienced appropriate ICD therapy patients was similar to patients who did not receive appropriate shocks (72.3 vs 70.7 years) and all SF-36 domains were similar without any significant differences or trends. The group who experienced inappropriate ICD therapy was small, but had a similar mean age and no significant differences or trends with

regard to any SF-36 domain. Furthermore, the group who required surgical intervention due to a complication had a similar mean age and SF-36 scores without significant differences.

In the primary prevention group, non-ischaemic vs ischaemic cardiomyopathy were compared. Notably, patients with an ischaemic aetiology were older (72.9 vs 66.9 years) but no SF-36 domain showed significantly different scores (RP showed a trend of lower score in ischaemic cardiomyopathy, p=0.067; ES=0.31).

## DISCUSSION

This Swedish cohort of general patients with ICD report worse HRQL than age-matched and sex-matched population norms. Except for BP and RE, all domains were

**Table 4** Subgroup analyses of patients with ICD

| SF-36 domains | Atrial fibrillation* P value | ES | Complications P value | ES | Appropriate therapy P value | ES | Inappropriate shock P value | ES |
|---|---|---|---|---|---|---|---|---|
| Physical functioning | **0.004*** | 0.41 | 0.500 | | 0.947 | | 0.592 | |
| Role physical | 0.128 | | 0.210 | | 0.897 | | 0.428 | |
| Bodily pain | **0.083*** | 0.27 | 0.167 | | 0.544 | | 0.381 | |
| General health | 0.606 | | 0.190 | | 0.863 | | 0.921 | |
| Vitality | **0.057*** | 0.31 | 0.321 | | 0.919 | | 0.765 | |
| Social functioning | 0.669 | | 0.910 | | 0.943 | | 0.944 | |
| Role emotional | 0.313 | | 0.972 | | 0.796 | | 0.955 | |
| Mental health | 0.912 | | 0.769 | | 0.747 | | 0.972 | |
| Physical component summary | **0.013*** | 0.38 | 0.107 | | 0.696 | | 0.464 | |
| Mental Component Summary | 0.923 | | 0.902 | | 0.772 | | 0.375 | |

Bold P values refer to significant values (<0.05).
*Effect sizes were lower when atrial fibrillation was present.
ES, effect size; ICD, implantable cardioverter defibrillator; SF-36, short form 36.

significantly lower although ESs were considered small. The component score, PCS and MCS, showed an equally small ES.

## Similar component score

In a recently published Danish study based on unselected ICD recipients, PCS (41.1 SD 10.3) and MCS (45.7 SD 10.8) at a mean follow-up of 7 years, were similar to our findings; poor PF, RP, GH and VT were associated with higher long-term mortality.[16] Early randomised controlled trials with shorter follow-up (1–3 years) reported similar findings.[17 18] In the Multicenter Automatic Defibrillator Implantation Trial II (MADIT II), PCS was associated with an 89% higher mortality, while MCS showed 39% higher mortality.[19] US data from the INTRINSIC RV trial found improvement of SF score over the first year and at 12 months, PCS was 42.2 and MCS 51.1, which is also similar to our results.[20]

## HF and HRQL

Systolic HF was the ICD indication for the vast majority of patients in our cohort, a third of whom received a CRT-D device. Several studies have determined that HF is a main determinant of poor HRQL.[21]

In a systematic review reviewing many studies based on SF-36 poor physical health status predicted adverse coronary artery disease and HF prognosis.[21] Using both disease-specific and generic test instruments, HF studies clearly revealed poor HRQL, but poor mental status had little impact. Moreover, follow-up time did not influence the predictive value of health status. In another review, those who reported poor disease-specific health status, in particular patients with HF, had increased mortality (HR 1.39) compared with those with moderate/good health status.[11] Systematic reviews in this field are hampered by severe heterogeneity across populations, different follow-up times, and diverse outcome measurements. Still, HF has repeatedly been shown to be a major determinant of poor health status.[12 22] Underlying cardiac disease and comorbidities influence HRQL and mortality.[12 23] A Cochrane review of non-ischaemic HF concluded: "using ICD therapy probably has little or no effect on quality of life."[24] This view is further strengthened by the fact that various patients with ICD without structural heart disease report similar SF-36 scores as the general populations.[25 26]

## Atrial fibrillation

A systematic review of several studies demonstrated poorer HRQL in patients with AF than healthy controls or general populations.[27] Notably, studies on AF and HRQL are diverse and often from specific cohorts as part of an evaluation of an intervention. Such studies are prone to selection bias because they likely reflect highly symptomatic patients. Nevertheless, more general patients with AF likewise show lower HRQL.[28 29] With a qualitative exploratory approach, the various symptoms have been further elucidated.[30] Patients with AF consist of subgroups including those with paroxysmal, persistent,

and permanent AF; and even within groups, the burden of symptoms may vary substantially. AF is common in patients with HF and thus prevalent in ICD cohorts. In our study, the comparison among those with/without a history of AF at baseline showed that the association of AF with poor HRQL was more pronounced for the physical domains, but this may be at least partly explained by older age.

## Age

As expected, age correlated with lower HQRL scores on physical domains. The four age strata showed that PCS deteriorated with older age and demonstrated significantly lower scores in the analyses. This is likely due to the burden of HF, AF and other comorbidities. Interestingly, MH domains were not affected and remained consistent throughout the age strata. It can be speculated that patients selected for ICD therapy reflect a group with relatively good MH status.

## Primary versus secondary indication

The proportion of primary prophylactic ICDs was low. This reflects a historical view on indication and possibly a more conservative approach towards primary prevention. Overall, the HRQL was similar and solely the GH domain was significantly lower in the primary prevention group. The coping strategy may differ between primary and secondary prevention patients. A survivor of a life-threatening arrhythmia may feel gratitude but be prone to anxiety and depression. A lower SF-36 score (all domains except BP) in primary vs secondary prevention has been shown, which underlines the vulnerability in the primary prevention group.[31] The differences may also be attributed to baseline characteristics; HF is a decisive factor for ICD implantation and translates into worse HRQL scores. Secondary prevention versus primary prevention patients are likely to have different expectations and acceptance of ICD therapy and its related complications.

## ICD shocks and complications

In the three separate subgroup analyses with regard to appropriate ICD therapy, inappropriate ICD shock and complications requiring surgery, no significant differences in the SF-36 domains were observed. This is somewhat reassuring, at least on a group level. Data from the INTRINSIC RV trial found no significant difference in HRQL between patients with and without ICD shocks.[20]

In a Danish cohort, HF was associated with poor scores on all SF-36 domains, while age and ICD shocks correlated less. Notably, on depression/anxiety-specific instruments, ICD shocks were a determinant for outcome. This suggests that it is the comorbidity burden, rather than the ICD itself, that determines the patients' HRQL and despite the fact that device-related complications can be distressing to individuals, ICDs are well tolerated overall. Of course, multiple ICD shocks may translate into worse HRQL, especially anxiety.[32] Our study was unable to detect a difference between patients with ICD shocks and those without. Several explanations have been suggested for this: lower incidence of shocks, ability to cope with complications over time,

generic instruments not specifically designed for ICDs, and the overall security and reassurance that comes from an ICD. Thus, it should be highlighted that careful device programming and management to avoid shocks and complications are crucial for optimal care.

## Gender perspectives

The vast majority (82%) of patients in our study were men. This is in line with a review of 19 studies, in which the percentage of men ranged from 54% to 83%. We found similar SF scores in men and women. In another study, only PF and VT were lower among women, although the difference was small, while for women, NYHA-class and Type D personality were strong determinants of poor HRQL.[23] Women may be more prone to anxiety and to score lower also on some SF-36 domains (PF, SF and MH) regardless of a history of ICD shock.[33] There needs to be an increased awareness of female candidates for ICD therapy, and our study support the fact that there are no overall sex differences with regard to generic HRQL.

## Clinical perspective

HRQL assessment has the potential to provide insights helpful to the overall management of patients with ICD, including selection of ICD candidates, and the reduction of hospitalisation and mortality.[34] Patients with ICDs have poorer general HRQL than the general population and increasing age and HF are strong determinants of worse HRQL. The underlying burden of comorbidities is the main driver for low HRQL rather than the ICD itself. Concerning quality-of-life improvement, overall management of the cardiac disease manifestations and comorbidities is important but challenging despite advances in many therapeutic fields. In fact, according to a meta-analysis there have been slight improvements regarding physical components, but not for mental components.[35] However, a comprehensive cardiac rehabilitation programme in patients with ICD improved both GH and MH.[34] Even though ICD shocks and complications can be devastating for an individual, the overall perception is that an ICD is generally well tolerated and does not affect generic HRQL. Individual assessment of benefits and risks, not gender category, should be part of the risk assessment strategy.

Furthermore, complications of ICD therapy do not have a significant impact on HRQL and should not be a reason to refrain from offering an ICD to otherwise eligible patients. Nevertheless, ICD is lifelong therapy and clinicians should regularly monitor the patient's ICD as well as quality of life as part of follow-up.

## Limitations

The major strength of our study is the use of a 'real-world' cohort, rather than strict inclusion/exclusion criteria used in randomised controlled trials, although this implies more heterogeneity than randomised controlled studies. The patients with ICD included in our study come from a cohort of unselected patients without referral centre bias. The SF-36 scores in the study group were compared with age-matched and sex-matched general population norms, that although more than 20 years old, likely still represent current status. The observational design reflects common practice outside a clinical trial, but may hamper by lack of details in assessment of clinical profiles even though medical records were scrutinised. The proportion who responded to the survey was comparatively high, which implies less risk of differential bias.

The HRQL assessment comes from a cross-sectional cohort studied long after initial implant and likely reflects views from a more stabilised phase of life as a patient with ICD. Nevertheless, the long-term impact on HRQL reported over the life of patients with ICD remains unknown. We lack information about personality traits and socioeconomic factors, which may affect self-reported health. SF-36 is a well-validated questionnaire that allows for comparisons on a group level but is not specifically developed for disease-specific assessment. The study design has limitations with respect to determining causal pathways and merely show associations. Therefore, it should be interpreted with caution.

## CONCLUSIONS

Patients with ICD have lower physical and mental HRQL, assessed by SF-36, than age-matched and sex-matched norms, although ESs are small. HRQL is similar regardless of sex, primary prevention or secondary prevention indication, appropriate therapy, inappropriate shock or complications but decreases with older age.

**Author affiliations**
[1]Institution of Medicine, Solna, Karolinska Institutet, Stockholm, Sweden
[2]Centre for Resarch and Development Region Gävleborg, Uppsala University, Gävle, Sweden
[3]Faculty of Health and Occupational Studies, Department of Caring Sciences, University of Gävle, Gävle, Sweden
[4]Faculty of Engineering and Sustainable Development; Department of Building, Energy and Sustainability Science, University of Gävle, Gävle, Sweden
[5]University Health Care Research Center, Faculty of Medicine and Health, Örebro University, Örebro, Sweden

**Acknowledgements** The authors acknowledge editing by Jo Ann LeQuang of LeQ Medical who reviewed the manuscript for American English use. Thanks to Kristina Bergström for her help with data collection through the questionnaires. Region Gävleborg funded this research project.

**Contributors** PM: design, collection and interpretation of data, statistical analyses, writing the article. GM: collection and interpretation of data, statistical analyses, critical revision writing the article. MW: interpretation of data, critical revision. JK: statistical analyses, interpretation of data, critical revision. All authors approved the manuscript for submission.

**Funding** The authors have not declared a specific grant for this research from any funding agency in the public, commercial or not-for-profit sectors.

**Competing interests** PM has received speaker fees or grants from Abbott, Alnylam, Amicus Therapeutics, Bayer, AstraZeneca, BMS, Boehringer-Ingelheim, Internetmedicin, Lilly, MSD, Novo Nordisk, Octopus Medical, Pfizer, Vifor Pharma and Zoll. GM has received speaker fee from Alnylam, Internetmedicin and MSD.

**Patient and public involvement** Patients and/or the public were not involved in the design, or conduct, or reporting, or dissemination plans of this research.

**Patient consent for publication** Not required.

**Ethics approval** The study complies with the Declaration of Helsinki and has been approved by the Ethical Review Board in Uppsala (document number 2018/416).

**Provenance and peer review** Not commissioned; externally peer reviewed.

**Data availability statement** All data relevant to the study are included in the article or uploaded as supplementary information. Data availability statement: All relevant data supporting the conclusions of this article is included within the article.

**Open access** This is an open access article distributed in accordance with the Creative Commons Attribution 4.0 Unported (CC BY 4.0) license, which permits others to copy, redistribute, remix, transform and build upon this work for any purpose, provided the original work is properly cited, a link to the licence is given, and indication of whether changes were made. See: https://creativecommons.org/licenses/by/4.0/.

**ORCID iD**
Peter Magnusson http://orcid.org/0000-0001-7906-7782

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
