## [Reviewer comments · BMJ Open]

ARTICLE DETAILS

TITLE (PROVISIONAL)	Health-related quality of life in patients with implantable cardioverter defibrillators in Sweden – a cross-sectional observational trial
AUTHORS	Magnusson, Peter; Mattsson, Gustav; Wallhagen, Marita; Karlsson, Jan

VERSION 1 – REVIEW

REVIEWER	Sears, Samuel F. East Carolina University, Psychology
REVIEW RETURNED	18-Dec-2020

GENERAL COMMENTS	Review: Health related quality of life in patients with implantable cardioverter defibrillators and: and observation trial The current study provides information on 223 patients recruited in Sweden over the course of a 10-year period from 2007 to 2017. This naturalistic sample evaluates QOL with a generic QOL instrument (SF 36) and compares the sample to normative samples. Results indicated that both physical and mental component summaries were significantly different between a normal sample and an ICD sample. Additional analysis examined the effects of shock and other key events and showed no differences. The authors conclude that ICD patients have lower physical and mental QOL than age and sex matched norms but the effect sizes are quite small. My primary reaction to this manuscript is that almost no “news” is presented here. The authors are to be congratulated for their long-term follow-up and focus on QOL. However, the manuscript struggles to provide anything new for the research literature. This type of QOL data (Schron et al.) has been available for almost 20 years. It's disappointing that there were not more measures of the ICD patient experience, such as cardiac anxiety, patient acceptance, ICD concerns, shocking anxiety, body image concerns and so forth. Unfortunately, generic QOL tools are notoriously insensitive in device studies. This could be added in the limitations. The most novel finding in the study is the role of atrial fibrillation being associated with poor quality of life. While that's well known in the broader literature, very little has been published about the ICD patient dealing with atrial fibrillation. Overall, the paper is reasonably well written, but fails to provide robust, new insights that would increase the enthusiasm for this paper.
---

	Introduction Good introduction to key concepts. May need to further strengthen the rationale for how this paper as an unselected cohort gives us a new perspective. Method Comparisons were reasonable. Wonder about shocks were less likely over time as improved programming from PainFree and Madit RIT were implemented during these years?
--	---

REVIEWER	Dessotte, Carina Universidade de São Paulo Escola de Enfermagem de Ribeirão Preto, Enfermagem Geral e Especializada
REVIEW RETURNED	29-Dec-2020

GENERAL COMMENTS	The article has the aim to determine health-related quality of life, using SF-36 questionnaire scores in unselected ICD patients in comparison with age- and sex-matched general Swedish population norms. The data were retrieved from electronic medical records (Melior™, Cerner Sverige AB, Stockholm). Data were collected of patients who had an ICD implanted or underwent device replacement between 1st January 2007 and 1st January 2017. Eligible patients were identified through Provisio™. It's a retrospective observational study. The following are some comments and requests for adjustments. Title: The title is very generic when compared to the aim of the study. By reading the title it is not possible for the reader to know that in addition to assessing the HRQoL of patients with ICD, they will also compare this measure with the HRQoL of people in the general population, according to sex and age. Adequacy is requested. Keyword: According to Mesh, "SF-36" is not a keyword. Limitations of this study Even after reading the article, it was not clear whether the data related to the assessment of HRQoL were obtained directly from the patients (sending the instruments to the patients in some way) or if they had already answered the instrument at some other time, and the authors only accessed this data. If it is the second case, because it is a collection of secondary data, it may be a limitation, because it was not designed to answer, specifically the question of this study. Introduction The introduction is clear and objective, presenting the problem to the reader. There is no justification for conducting the study: are there no such studies in the literature? Is the data that exists divergent? The elaboration of the justification for conducting the study is requested. Method Data were collected from patients who had an ICD implanted or underwent device replacement between 1st January 2007 and 1st January 2017. The authors did not describe the study's temporal design in the method. Upon reaching the results, we observed that the data were collected a single moment in time, so we ask for the
---

	description that this is a cross-sectional study. This information must be in the method. The authors did not describe the sample inclusion and exclusion criteria. At the end of the discussion, the authors “justify” the absence of these criteria, as, according to them, it is a real Cohort. It should be noted that even in the cohorts, we must reflect on the inclusion and exclusion criteria. For example, can a patient who has an ICD and a resynchronizer be evaluated in the same way? Or a patient with an ICD and pacemaker? From what age were the patients inserted (an adolescent may have a very different assessment from that of an adult)? How long have the patients been with the ICD (we already have available in the literature that the first post-implant year is the most challenging)? Because it is an assessment of subjective constructs, even that person with an ICD without cognitive conditions is able to participate? Adequacy is requested. It is questioned why the data collection included patients who implanted the ICD between 1st January 2007 and 1st January 2017. It is questioned how long ago the patient had implanted the ICD at the time of collection, since we already have evidence in the literature that the first year is the most aversive and psychologically challenging due to the various necessary adaptations. Thus, the implantation time can be a confusing variable for the results presented here. What are the criteria adopted for the classification of age in: 32-59 years, 60-69 years, 70-79 years, and ≥ 80 years? It is not described in the method how the authors matched age, sex and scores of the general population with these same variables of patients with ICD, for the proposed comparisons. Clarifications are requested. When analyzing the results, it is observed that the patients with ICD were mostly older people and male. Do the scores used by the general population also reflect this characterization? In data analysis, the authors did not describe which variables they were associating, thus linking with the chosen test. Clarifications are requested. The correct way to describe the SPSS software is: IBM-SPSS program, version 22.0 for Windows (or Apple) (SPSS, Inc., Chicago, IL, USA). In the method, the authors stated that “Patients were not involved in the design, or conduct, or reporting, or dissemination plan of our research”, however, in the first sentence of the results we found the following statement: “The SF-36 questionnaire was returned by 223 ICD patients which corresponds to a response rate of 77.2%”. Thus, it is questioned and requested to adapt the method on the collection of data related to HRQoL. Did participants already
--	---

	complete SF-36 (if so, at what time) during hospitalization or were the instruments sent to participants and how were these instruments sent? Results As in the method the authors did not describe which variables they were associating, according to the chosen test, when we arrived at the results we identified that the correlation between age (considering a numeric variable) and the eight SF-36 domains was performed, as well as the association of the range age (four categories) with the eight SF-36 domains. The reason for having performed two tests for the same objective is questioned. Regarding the performance of the Kruskal-Wallis test, which investigated the interference of age in the eight domains of the SF-36, it is requested for those results that were statistically significant to perform the Bonferroni multiple comparison test, to identify between which categories age is the difference in each domain.
--	--

VERSION 1 – AUTHOR RESPONSE

Reviewer 1	We are thankful for this constructive review. It will definitely improve the paper.
#2. The authors are to be congratulated for their long-term follow-up and focus on QOL. However, the manuscript struggles to provide anything new for the research literature. This type of QOL data (Schron et al.) has been available for almost 20 years. It's disappointing that there were not more measures of the ICD patient experience, such as cardiac anxiety, patient acceptance, ICD concerns, shocking anxiety, body image concerns and so forth. Unfortunately, generic QOL tools are notoriously insensitive in device studies. This could be added in the limitations. The most novel finding in the study is the role of atrial fibrillation being associated with poor quality of life. While that's well known in the broader literature, very little has been published about the ICD patient dealing with atrial fibrillation. Overall, the paper is reasonably well written, but fails to provide robust, new	Thank you for recognizing the novelty regarding the association between poor quality of life and atrial fibrillation in this population. We do admit the limitation you addressed and have further reinforced it in the bullet points:  • Lack of ICD-specific patient-reported questions. Page 3, line 8.

insights that would increase the enthusiasm for this paper.	
#3. Introduction Good introduction to key concepts. May need to further strengthen the rationale for how this paper as an unselected cohort gives us a new perspective.	Thanks for the positive wording about key concepts. We have stressed the importance of unselected patients outside clinical trials because this is needed in order to generalize findings. “...it remains largely unknown if this can be generalized into unselected ICD cohorts with long-term follow-up outside of a clinical trial.^{11,12} Therefore, the primary aim of this study was to determine SF-36 scores in unselected ICD patients in comparison with age- and sex-matched general Swedish population norms. Page 5, line 1-5.
#4. Method Comparisons were reasonable. Wonder about shocks were less likely over time as improved programming from PainFree and Madit RIT were implemented during these years?	It can be speculated if, and to what extent, implementation of tailored-programming and the findings from the mentioned trials have been translated into clinical practice. This has been addressed in a previous publication: https://onlinelibrary.wiley.com/doi/full/10.1111/pace.13869
Reviewer 2	We are thankful for this constructive review. It definitely improved the paper.
#5. The title is very generic when compared to the aim of the study. By reading the title it is not possible for the reader to know that in addition to assessing the HRQoL of patients with ICD, they will also compare this measure with the HRQoL of people in the general population, according to sex and age. Adequacy is requested.	The title reflects the content. If we would add “age- and sex-matched population norms” it would be too long considering the requirement to include study type. If the Editor suggests this we will do it. Page 1, line 1-2.
#6. Keyword: According to Mesh, “SF-36” is not a keyword.	Agree. We deleted “SF-36” as a key word.
#7. Limitations of this study Even after reading the article, it was not clear whether the data related to the assessment of HRQoL were obtained directly from the patients (sending the instruments to the patients in some way) or if they had already answered the instrument at some other time, and the authors only accessed this data. If it is the second case, because it is a collection of secondary data, it may be a limitation, because it was not	Agree. We have clarified this in the Method section: “The questionnaires were distributed by regular mail, including two reminders, during 2019.” Page 5, line 20.

designed to answer, specifically the question of this study.	
#8. Introduction The introduction is clear and objective, presenting the problem to the reader. There is no justification for conducting the study: are there no such studies in the literature? Is the data that exists divergent? The elaboration of the justification for conducting the study is requested.	Thanks for the positive view on the Introduction. We have stressed the importance of unselected patients outside clinical trials because this is needed in order to generalize findings. “...it remains largely unknown if this can be generalized into unselected ICD cohorts with long-term follow-up outside of a clinical trial.^{11,12} Therefore, the primary aim of this study was to determine SF-36 scores in unselected ICD patients in comparison with age- and sex-matched general Swedish population norms. Page 5, line 1-5.
#9. Method Data were collected from patients who had an ICD implanted or underwent device replacement between 1st January 2007 and 1st January 2017. The authors did not describe the study's temporal design in the method. Upon reaching the results, we observed that the data were collected a single moment in time, so we ask for the description that this is a cross-sectional study. This information must be in the method.	Agree. See #7.
#10. The authors did not describe the sample inclusion and exclusion criteria. At the end of the discussion, the authors “justify” the absence of these criteria, as, according to them, it is a real Cohort. It should be noted that even in the cohorts, we must reflect on the inclusion and exclusion criteria. For example, can a patient who has an ICD and a resynchronizer be evaluated in the same way? Or a patient with an ICD and pacemaker? From what age were the patients inserted (an adolescent may have a very different assessment from that of an adult)? How long have the patients been with the ICD (we already have available in the literature that the first post-implant year is the most challenging)? Because it is an assessment of subjective constructs, even that person with an ICD without cognitive	The rationale of the study was assess HRQL in broad, unselected sample of ICD patients including CRTD-patients, but not pacemaker patients. In Table 1 the cohort is described including the mean age at implant and mean age at the time of evaluation with SF-36. We solely included adult patients. This has been clarified in the following sentence: “...observational study covered adult patients...” Page 5, line 10. Indeed, all patients were sent the questionnaire independent of cognitive status, which was unknown to the researchers.

conditions is able to participate? Adequacy is requested.	
#11. It is questioned why the data collection included patients who implanted the ICD between 1st January 2007 and 1st January 2017.	The time period was arbitrarily defined as 10 year, although justified by sample size, long-term assessment of outcome (rather than first year after implant), and similar management of patients and selection criteria for ICD.
#12. It is questioned how long ago the patient had implanted the ICD at the time of collection, since we already have evidence in the literature that the first year is the most aversive and psychologically challenging due to the various necessary adaptations. Thus, the implantation time can be a confusing variable for the results presented here.	This is addressed in Table 1, the mean age at implant and distribution of the SF-36 are reported. Mean age at implant: 64.1 ± 10.9 years Mean age at SF-36: 71.1 ± 9.7 years See #7.
#13. What are the criteria adopted for the classification of age in: 32-59 years, 60-69 years, 70-79 years, and ≥ 80 years?	These age strata is often used and were pre-defined.
#14. It is not described in the method how the authors matched age, sex and scores of the general population with these same variables of patients with ICD, for the proposed comparisons. Clarifications are requested. When analyzing the results, it is observed that the patients with ICD were mostly older people and male. Do the scores used by the general population also reflect this characterization?	This is described in the Method section: The SF-36 profile in the study group was compared to a general population sample which was randomly selected from the Swedish SF-36 normative database (n=8,930; response rate 68%). ¹⁵ The normative sample (validated in Sweden 1991-92) was matched on sex and age and comprised 171 persons (141 males) with a mean age of 68.8 years (SD 11.1).
#15. In data analysis, the authors did not describe which variables they were associating, thus linking with the chosen test. Clarifications are requested. The correct way to describe the SPSS software is: IBM-SPSS program, version 22.0 for Windows (or Apple) (SPSS, Inc., Chicago, IL, USA).	This is standard reporting and explained in the Method section. Data were described as frequencies, percentages, means including standard deviations (SD), and 95% confidence intervals (CIs) were used. The t-test was used for comparisons of continuous variables. Differences in SF-36 domains between groups were tested from the non-parametric Mann-Whitney U-test. Thanks for the reference format of SPSS. We checked this on the official IBM website and revised accordingly: "IBM SPSS Statistics for Windows, Version 22.0. Armonk, NY: IBM Corp." https://www.ibm.com/support/pages/how-cite-ibm-spss-statistics-or-earlier-versions-spss

#16. In the method, the authors stated that “Patients were not involved in the design, or conduct, or reporting, or dissemination plan of our research”, however, in the first sentence of the results we found the following statement: “The SF-36 questionnaire was returned by 223 ICD patients which corresponds to a response rate of 77.2%”. Thus, it is questioned and requested to adapt the method on the collection of data related to HRQoL. Did participants already complete SF-36 (if so, at what time) during hospitalization or were the instruments sent to participants and how were these instruments sent?	See #7. The statement “Patient and public involvement Patients were not involved in the design, or conduct, or reporting, or dissemination plan of our research.” refers to the research methodology and data acquisition from the researcher’s perspective. This sentence is required by the journal.
#17. As in the method the authors did not describe which variables they were associating, according to the chosen test, when we arrived at the results we identified that the correlation between age (considering a numeric variable) and the eight SF-36 domains was performed, as well as the association of the range age (four categories) with the eight SF-36 domains. The reason for having performed two tests for the same objective is questioned. Regarding the performance of the Kruskal-Wallis test, which investigated the interference of age in the eight domains of the SF-36, it is requested for those results that were statistically significant to perform the Bonferroni multiple comparison test, to identify between which categories age is the difference in each domain.	See #15. Kruskal-Wallis test is standard reporting in comparison of age data in the field.

VERSION 2 – REVIEW

REVIEWER	Dessotte, Carina Universidade de São Paulo Escola de Enfermagem de Ribeirão Preto, Enfermagem Geral e Especializada
REVIEW RETURNED	16-Mar-2021

GENERAL COMMENTS	As asked previoully, the authors did not described the sample inclusion and exclusion criteria. They claim this results are a part
--

	of another research, but, it is necessary here be be clean what were the sample inclusion and exclusion criteria. When reading an article, we should find all the information in it, and there is no need to go back to another article to identify the gaps presented. Therefore, they did not respond to this pending issue. In response to a pending issue, the authors stated that the SF-36 questionnaire was sent by email, during the year 2019. They did not answer the reason for the time frame: they included patients who implanted an ICD in the period between 1st January 2007 and 1st January 2017. Once again, regarding the performance of the Kruskal-Wallis test, which investigated the interference of age in the eight domains of the SF-36, it is requested for those results that were statistically significant to perform the Bonferroni multiple comparison test, to identify between which categories age is the difference in each domain.
--	---

VERSION 2 – AUTHOR RESPONSE

Reviewer 2	We are thankful for this constructive review.
#2. As asked previously, the authors did not described the sample inclusion and exclusion criteria. They claim this results are a part of another research, but, it is necessary here be clean what were the sample inclusion and exclusion criteria. When reading an article, we should find all the information in it, and there is no need to go back to another article to identify the gaps presented. Therefore, they did not respond to this pending issue. In response to a pending issue, the authors stated that the SF-36 questionnaire was sent by email, during the year 2019. They did not answer the reason for the time frame: they included patients who implanted an ICD in the period between 1st January 2007 and 1st January 2017. Once again, regarding the performance of the Kruskal-Wallis test, which investigated the interference of age in the eight domains of the SF-36, it is requested for those results that were statistically significant to	The inclusion criteria are stated in the Method section as follows: This retrospective observational study covered adult patients in Region Gävleborg, Sweden, who had an ICD implanted or underwent device replacement between 1st January 2007 and 1st January 2017. All relevant data, including Table 1 are reported in the current paper; there is no need to go to the paper published about appropriate therapy, complications, and mortality, but it is referred to for clarification. As mentioned in the Method: “The questionnaires were distributed by regular mail, including two reminders, during 2019.” Thus, no questionnaires were sent by email. The time period was arbitrarily defined as 10 year, although justified by sample size, long-term assessment of outcome (rather than first year after implant), and similar management of patients and selection criteria for ICD. Kruskal-Wallis test is standard reporting in comparison of age data in the field. Age was categorized into the four strata: 32-59 years, 60-69 years, 70-79 years, and ≥80 years and differences between age strata were analyzed using the Kruskal-Wallis non-parametric analysis of variance. The associations between age and HQRL were tested using Spearman's non-parametric correlation

perform the Bonferroni multiple comparison test, to identify between which categories age is the difference in each domain.	coefficient for analyses and Pearson’s correlation analysis to confirm results. Indeed, these statistical tests confirmed each others. We judged that Bonferroni multiple comparison test would not add essential information in this regard and would increase the risk of type II errors. Regarding which categories are different in each domain significant in the Kruskal-Wallis we think that the data is easily interpreted as is with worsening scores with increasing age. For example the mean score for the domain Physical Functioning is 79.4, 71.9, 62.7, and 51.4 for the respective age categories. The same is valid for the domain Role Physical. The summary score PCS is built up on mainly of these domain.
--	---